# Predicting physiologically-relevant oxygen concentrations in precision-cut liver slices using mathematical modelling

S. J. Chidlow[1]*, L. E. Randle[2�} R. A. Kelly[1,3�}

**1** School of Computer Science and Mathematics, Liverpool John Moores University, Liverpool, United Kingdom, **2** School of Pharmacy and Biomolecular Science, Liverpool John Moores University, Liverpool, United Kingdom, **3** Syngenta, Early-Stage Research, Jeallot's Hill Research Centre, Bracknell, United Kingdom

}These authors contributed equally to this work.
\* S.J.Chidlow@ljmu.ac.uk

**Data Availability Statement:** All relevant data are within the paper and its Supporting information files.

## Abstract

Precision cut liver slices represent an encouraging *ex vivo* method to understand the pathogenesis of liver disease alongside drug induced liver injury. Despite being more physiologically relevant compared to *in vitro* models, precision cut liver slices are limited by the availability of healthy human tissue and experimental variability. Internal oxygen concentration and media composition govern the longevity and viability of the slices during the culture period and as such, a variety of approaches have been taken to maximise the appropriateness of the internal oxygen concentrations across the slice. The aim of this study was to predict whether it is possible to generate a physiologically relevant oxygen gradient of 35-65mmHg across a precision cut liver slice using mathematical modelling. Simulations explore how the internal oxygen concentration changes as a function of the diameter of the slice, the position inside the well and the external incubator oxygen concentration. The model predicts that the desired oxygen gradient may be achieved using a 5mm diameter slice at atmospheric oxygen concentrations, provided that the slice is positioned at a certain height within the well of a 12-well plate.

## Introduction

Understanding the pathogenesis of liver diseases such as liver fibrosis, cancer and hepatitis alongside drug-induced liver injury (DILI) requires physiologically-relevant experimental models. Current *in vitro* models, while reliable, fall short when predicting liver injury and disease as they lack the complex cellular interactions that are present within the native tissue environment [1]. In primary cells or cell lines, the diminishing communication between the extracellular environment, amongst other factors, causes de-differentiation [2]. 2D models focussing on a single cell type are now considered too simplistic while in recent years, three-dimensional (3D) models e.g spheroids and organoids have gained popularity. However, despite the inclusion of multiple cell types, they cannot fully recapitulate the *in vivo* setting [3].

**Funding:** The authors received no specific funding for this work.

**Competing interests:** The authors have declared that no competing interests exist.

Alternative *in vivo* approaches are also flawed. The large number of animal models required to conduct such studies raises legitimate ethical questions, with the field of drug discovery actively reducing, replacing and refining the use of animal models in general. Scientifically, *in vivo* models are also limited by species differences making it difficult to capture molecular pathogenesis [2].

Precision cut liver slices (PCLS) represent a promising *ex vivo* alternative, possessing the potential to mitigate the aforementioned *in vitro* and *in vivo* shortcomings in liver disease pathogenesis [4]. PCLS models have been deployed since the 1980's, first described by Smith et al. [5] after Carlos Krumdieck described the fast-production Krumdieck tissue slicer [6]. Machine-cut slices are highly reproducible and reduce the number of animal models used experimentally. With respect to their functionality, PCLS models better represent the *in vivo* environment compared to classical *in vitro* methods, recapitulating the complex multicellular histoarchitecture and maintaining complex biochemical and molecular processes [2].

Like any experimental systems, the PCLS method is not without limitations. Access to 'healthy' human tissue is the predominant limitation, with variability originating from differences in the heath of human sample sources and only providing a snapshot of the infiltrating immune cell population. Furthermore, viability of the PCLS culture is also a limitation. Slices can be cultured for up to 5 days [7], with reports of extending this to 15 days under certain conditions, however, such culture periods may be unsuitable to study chronic effects [8].

Experimental variability has decreased since the publication of improved protocols, particularly by de Graaf et. al. [7]. The ultimate aim when using this *ex vivo* method is to construct a reproducible and viable model. Model viability depends on a number of factors, including the dimensions of the PCLS (diameter and thickness), internal oxygenation and media composition [7, 9], which are usually determined emperically. These factors influence the ease of access of nutrients and oxygen to the inner cell layers, which in turn can influence the longevity of the PCLS culture period.

One of the experimental options is the thickness of the PCLS. The typical thickness of a slice ranges from 100 to 300$\mu m$, [10], with 300$\mu m$ considered to be the maximum thickness that allows a full and complete supply of oxygen and nutrients to reach the inner cells [11]. As such, it is commonplace to use a thickness of approximately 250$\mu m$. Another experimental option is controlling external oxygen concentration in the incubator, to allow sufficient diffusion of oxygen across the tissue. Different approaches may be taken, for example, some experiments rely on elevated external oxygen concentration in the region of 80–95% oxygen [7], while others use a combination of atmospheric oxygen concentration at 21% and a trans-well insert and platform rocking [8, 9]. Both approaches share the same endpoint; to maximise the likelihood of appropriate internal oxygen concentrations across the PCLS.

Zonation (spatial heterogeneity) occurs within physiological environments providing gradients in terms of enzymes, substrates, oxygen, nutrients etc. Each tissue in the body possesses a unique balance of gasses as a function of vascularization [12]. The liver is well-known to have an oxygen gradient, specifically 60–65mmHg in the periportal blood, to 30–35mmHg in the perivenous blood [13]. The ability to recapitulate this gradient across a PCLS would greatly enhance the physiological-relevance of the system.

Human tissue availability and experimental costs preclude extensive experimental analysis of these various factors. Various digital methods could be employed e.g machine learning, deep learning and artificial intelligence to aid in the optimisation of the PCLS experimental conditions, however these systems require much larger data sets to train and validate the platform algorithms. Such methods are increasingly being used in digital pathology and risk stratification offering unbiased and automated classification [14].

The aim of this study is to determine whether it is possible to experimentally generate a physiologically-relevant oxygen gradient of 35–65mmHg across a PCLS using a mathematical model, to inform our future wet lab experiments. The model explores the effects of i) three external incubator oxygen concentrations (21, 80 and 95%), ii) two PCLS diameter lengths (5mm and 8mm) and iii) the position of the slice inside the well.

## Materials and methods

### Model formulation

The steady state concentration of oxygen in a PCLS was considered in this study. The tissue is assumed to be circular in shape and suspended in fluid inside one well of a 12-well plate. It is assumed that the fluid and tissue together occupy the region

$$0 < r < r_o, \qquad 0 < z < H,$$

where $r_o$ is the radius of the the well and $H$ is the depth of the fluid. The tissue itself has radius $r_T$ and occupies the region $0 < r < r_T$, $h_1 < z < h_2$. The fluid occupies the rest of the domain. A definition sketch of the problem is given in Fig 1.

The concentration of oxygen within the fluid is denoted $\phi_1$ and satisfies the steady-state diffusion equation:

$$\frac{\partial^2 \phi_1}{\partial r^2} + \frac{1}{r}\frac{\partial \phi_1}{\partial r} + \frac{\partial^2 \phi_1}{\partial z^2} = 0. \tag{1}$$

As oxygen will both diffuse and be absorbed by the liver slice, it is assumed that oxygen concentration within the tissue (denoted $\phi_2$) satisfies the following PDE

$$D_2 \left( \frac{\partial^2 \phi_2}{\partial r^2} + \frac{1}{r}\frac{\partial \phi_2}{\partial r} + \frac{\partial^2 \phi_2}{\partial z^2} \right) - V\phi_2 = 0, \tag{2}$$

so that $V$ defines the rate at which oxygen is absorbed by the tissue and $D_2$ denotes the diffusion coefficient of the liver slice.

In order to fully describe the mathematical problem, boundary and matching conditions were applied to the fluid and tissue. The oxygen concentration at the surface of the fluid is assumed to be known and is denoted $\phi_0$. It is also assumed that the well is perfectly insulated

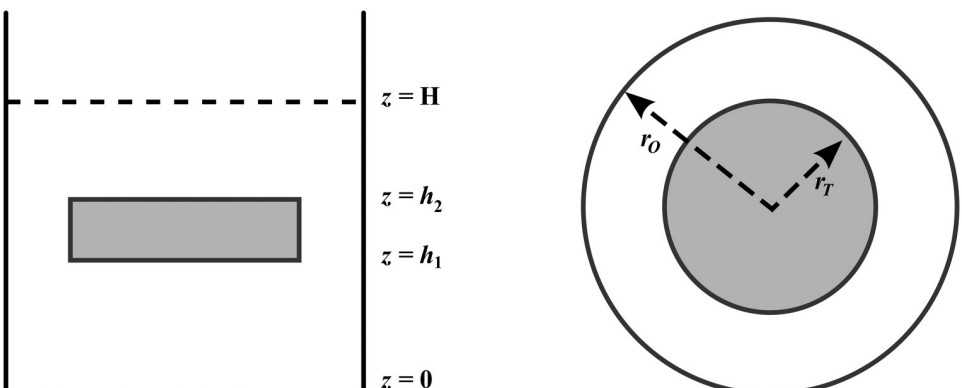

**Fig 1. Schematic of the PCLS system.** The setup can be visualised side-on (left) or top-down (right). $H$ represents the depth of the media, while $h_1$ and $h_2$ represent the bottom and top of the slice respectively. The radius of the well and the radius of the tissue slice are denoted $r_O$ and $r_T$ respectively. The total volume of media per well is 1.3ml.

so that there are "no-flow" conditions on the sides and bottom of the well. Mathematically, these boundary conditions can be expressed as follows:

$$\phi(r, H) \quad = \phi_0, \qquad 0 < r < r_o, \tag{3}$$

$$\frac{\partial \phi}{\partial r}(r_o, z) \quad = 0, \qquad 0 < z < H, \tag{4}$$

$$\frac{\partial \phi}{\partial z}(r, 0) \quad = 0, \qquad 0 < r < r_o. \tag{5}$$

Ensuring that the oxygen concentration and mass flux are continuous at the interface between the fluid and the liver slice, the following matching conditions are observed:

$$\phi_1(r, h_i) \quad = \phi_2(r, h_i), \qquad 0 < r < r_T, \tag{6}$$

$$D_1 \frac{\partial \phi_1}{\partial r}(r, h_i) \quad = D_2 \frac{\partial \phi_2}{\partial r}(r, h_i), \qquad 0 < r < r_T, \tag{7}$$

where $i = 1, 2$, and

$$\phi_1(r_T, z) \quad = \phi_2(r_T, z), \qquad h_1 < z < h_2, \tag{8}$$

$$D_1 \frac{\partial \phi_1}{\partial r} \quad = D_2 \frac{\partial \phi_2}{\partial r}, \qquad h_1 < z < h_2. \tag{9}$$

Note that $D_1$ denotes the diffusion coefficient within the media.

## Method of solution

Rather than solve for the oxygen concentrations in the liver slice and tissue separately, the overall domain was split into two separate regions. The outer region, denoted region 1, occupies the domain $r_T < r < r_o$, $0 < z < H$ and consists entirely of fluid. The inner region, denoted region 2, contains both the liver slice and fluid.

### The inner region

The oxygen concentration in the inner region, denoted $\phi_{II}$ satisfies the following equations:

$$\frac{\partial^2 \phi_{II}}{\partial r^2} + \frac{1}{r} \frac{\partial \phi_{II}}{\partial r} + \frac{\partial^2 \phi_{II}}{\partial z^2} = 0, \qquad z \in (0, h_1) \cup (h_2, H), \tag{10}$$

$$\frac{\partial^2 \phi_{II}}{\partial r^2} + \frac{1}{r} \frac{\partial \phi_{II}}{\partial r} + \frac{\partial^2 \phi_{II}}{\partial z^2} - \frac{V}{D_2} \phi_{II} = 0, \qquad h_1 \leq z \leq h_2, \tag{11}$$

subject to the following boundary and matching conditions

$$\frac{\partial \phi_{II}}{\partial z}(r, 0) = 0, \tag{12}$$

$$\phi_{II}(r, h_1^-) = \phi_{II}(r, h_1^+), \tag{13}$$

$$D_1 \frac{\partial \phi_{II}}{\partial z}(r, h_1^-) = D_2 \frac{\partial \phi_{II}}{\partial z}(r, h_1^+), \tag{14}$$

$$\phi_{II}(r, h_2^-) = \phi_{II}(r, h_2^+), \tag{15}$$

$$D_2 \frac{\partial \phi_{II}}{\partial z}(r, h_2^-) = D_1 \frac{\partial \phi_{II}}{\partial z}(r, h_2^+), \tag{16}$$

$$\phi_{II}(r, H) = \phi_0. \tag{17}$$

In order to solve these equations, solutions in the form

$$\phi_{II}(r, z) = U_{II}(z) + \psi_{II}(r, z), \tag{18}$$

were sought, where $U_{II}$ satisfies Eqs (10)–(17) exactly and $\psi_{II}(r, z)$ satisfies the corresponding homogeneous versions of (10)–(17). It can be verified that

$$U_{II}(z) = \begin{cases} \dfrac{D_1 \phi_0}{\theta}, & 0 \leq z \leq h_1, \\[2mm] \dfrac{D_1 \phi_0}{\theta} \cosh(k(z - h_1)), & h_1 < z < h_2, \\[2mm] \phi_0 \left( 1 + \dfrac{D_2 k}{\theta} \sinh(k(h_2 - h_1))(z - H) \right), & h_2 \leq z \leq H, \end{cases} \tag{19}$$

where

$$k = \sqrt{\frac{V}{D_2}},$$

$$\theta = D_1 \cosh(k(h_2 - h_1)) + D_2 k(H - h_2)\sinh(k(h_2 - h_1)).$$

The solution of $\psi_{II}(r, z)$ may be found by looking for separable solutions of the form

$$\psi_{II}(r, z) = R(r)Z(z). \tag{20}$$

Substituting this expression into Eqs (10)–(17) yields the equation

$$r^2 \frac{d^2 R}{dr^2} + r \frac{dR}{dr} - r^2 \lambda R = 0, \tag{21}$$

together with the following eigenvalue problem

$$Z'' + \lambda Z = 0, \qquad z \in (0, h_1) \cup (h_2, H), \tag{22}$$

$$Z'' + (\lambda - k^2)Z = 0, \qquad h_1 < z < h_2, \tag{23}$$

$$Z'(0) = 0, \tag{24}$$

$$Z(h_1^-) = Z(h_1^+), \tag{25}$$

$$D_1 Z'(h_1^-) = D_2 Z'(h_1^+), \tag{26}$$

$$Z(h_2^-) = Z(h_2^+), \tag{27}$$

$$D_2 Z'(h_1^-) = D_1 Z'(h_1^+), \tag{28}$$

$$Z(H) = 0. \tag{29}$$

This problem possesses only positive eigenvalues ($\lambda > 0$), with $N \geq 1$ eigenvalues in the range $0 < \sqrt{\lambda} = \eta < k$ that satisfy the equation

$$
\begin{aligned}
1 + \tan(\eta h_1)\tan(\eta(h_2 - H)) \quad &+ \frac{D_2 \tau}{D_1 \eta}\tanh(\tau(h_2 - h_1))\tan(\eta(h_2 - H)) \\
&+ \frac{D_1 \eta}{D_2 \tau}\tan(\eta h_1)\tanh(\tau(h_1 - h_2)) = 0,
\end{aligned}
\tag{30}
$$

where $\tau = \sqrt{k^2 - \eta^2}$. The corresponding eigenfunctions are

$$
Z_n^{(2)}(z) =
\begin{cases}
\cos(\eta_n z), & 0 \leq z < h_1, \\[2mm]
\cos(\eta_n h_1)\cosh(\tau_n(z - h_1)) - \dfrac{D_1 \eta_n}{D_2 \tau_n}\sin(\eta_n h_1)\sinh(\tau_n(z - h_1)), & h_1 \leq z \leq h_2, \\[2mm]
\rho_n \sin(\eta_n(z - H)), & h_2 < z \leq H,
\end{cases}
\tag{31}
$$

where

$$
\rho_n = \frac{1}{\sin(\eta_n(h_2 - H))}\left( \cosh(\tau_n(h_1 - h_2))\cos(\eta_n h_1) + \frac{D_1 \eta_n}{D_2 \tau_n}\sinh(\tau_n(h_1 - h_2))\sin(\eta_n h_1) \right).
$$

and $n = 1, 2, \ldots, N$.

The remaining eigenvalues $k < \sqrt{\lambda} = \mu$ satisfy the equation

$$
\begin{aligned}
\frac{D_1 \mu}{D_2 \delta}\tan(\delta(h_2 - h_1))\tan(\mu h_1) \quad &- \frac{D_2 \delta}{D_1 \mu}\tan(\mu(h_2 - H))\tan(\delta(h_2 - h_1)) \\
&= 1 + \tan(\mu(h_2 - H))\tan(\mu h_1),
\end{aligned}
\tag{32}
$$

where

$$\delta = \sqrt{\mu^2 - k^2}.$$

The corresponding eigenfunctions are

$$
Zn^{(2)}(z) = \begin{cases} \cos(\mu_n z), & 0 \leq z < h_1, \\[2mm] \cos(\mu_n h_1)\cos(\delta_n(z-h_1)) - \dfrac{D_1\mu_n}{D_2\delta_n}\sin(\mu_n h_1)\sin(\delta_n(z-h_1)), & h_1 \leq z \leq h_2, \\[2mm] \zeta_n \sin(\mu_n(z-H)), & h_2 < z \leq H, \end{cases} \tag{33}
$$

where

$$
\zeta_n = \frac{1}{\sin(\mu_n(H-h_2))}\left( \frac{D_1\mu_n}{D_2\delta_n}\sin(\delta_n(h_2-h_1))\sin(\mu_n h_1) - \cos(\delta_n(h_2-h_1))\cos(\mu_n h_1) \right)
$$

and $n \geq N$.

Given that all of the eigenvalues are positive, the solutions of (21) are

$$
R_n(r) = \alpha_n I_0(\sqrt{\lambda_n}r) + \beta_n K_0(\sqrt{\lambda_n}r), \tag{34}
$$

where $I_0$ and $K_0$ denote the modified Bessel functions of order zero. Using (20) and (18), we can form the solution

$$
\phi_{II}(r,z) = U_{II}(z) + \sum_{n=1}^{\infty} \alpha_n I_0(\sqrt{\lambda_n}r)Z_n^{(2)}(z), \tag{35}
$$

which holds in the region $0 \leq r < r_T$. Note that $\beta_n = 0$, $n \in \mathbb{N}$ to ensure that the solution remains bounded as $r \to 0$.

## The outer region

As this region contains only fluid, the oxygen concentration ($\phi_I$) satisfies the following equations

$$
\frac{\partial^2 \phi_I}{\partial r^2} + \frac{1}{r}\frac{\partial \phi_I}{\partial r} + \frac{\partial^2 \phi_I}{\partial z^2} = 0, \qquad 0 < z < H, \tag{36}
$$

$$
\frac{\partial \phi_I}{\partial z}(r,0) = 0, \tag{37}
$$

$$
\phi_I(r,H) = \phi_0. \tag{38}
$$

Following the same approach as in the inner region, the solutions in the following form were sought

$$
\phi_I(r,z) = U_I(z) + \psi(r,z). \tag{39}
$$

Substituting this expression into (36)–(38) and looking for separable solutions of $\psi$ in the same form as (20) reveals the eigenvalue problem

$$
Z'' + \lambda Z = 0, \qquad 0 < z < H, \tag{40}
$$

$$
Z'(0) = 0, \tag{41}
$$

$$
Z(H) = 0. \tag{42}
$$

It may be easily determined that this problem possesses the eigenvalues

$$\lambda_n = \omega_n^2 = \left(\left(n - \frac{1}{2}\right)\pi\right)^2, \tag{43}$$

and corresponding eigenfunctions

$$Z_n^{(1)}(z) = \cos(\omega_n z). \tag{44}$$

It then follows that

$$R_n(r) = A_n I_0(\omega_n r) + B_n K_0(\omega_n r). \tag{45}$$

As the function $U_I(z) = \phi_0$, the following solution may be formed

$$\phi_I(r, z) = \phi_0 + \sum_{n=1}^{\infty} (A_n I_0(\omega_n r) + B_n K_0(\omega_n r))\cos(\omega_n z). \tag{46}$$

This solution must satisfy the no-flow condition

$$\frac{\partial \phi_I}{\partial r}(ro, z) = 0,$$

which further yields the relationship

$$B_n = -\frac{I_0'(\omega_n ro)A_n}{K_0'(\omega_n ro)},$$

where $'$ denotes differentiation with respect to $r$ and $n \in \mathbb{N}$. The final form of the outer solution is then

$$\phi_I(r, z) = \phi_0 + \sum_{n=1}^{\infty} A_n(K_0'(\omega_n ro)I_0(\omega_n r) - I_0'(\omega_n ro)K_0(\omega_n r))\cos(\omega_n z). \tag{47}$$

## Matching inner and outer solutions

In order to find the constants $\alpha_n$ and $A_n$, the following matching conditions are applied at the boundary $r = r_T$

$$\phi_I(r_T, z) = \phi_{II}(r_T, z), \qquad 0 < z < H, \tag{48}$$

$$\frac{\partial \phi_I}{\partial r}(r_T, z) = \frac{\partial \phi_{II}}{\partial r}(r_T, z), \qquad 0 < z < h_1 \ \cup \ h_2 < z < H, \tag{49}$$

$$D_1 \frac{\partial \phi_I}{\partial r}(r_T, z) = D_2 \frac{\partial \phi_{II}}{\partial r}(r_T, z), \qquad h_1 < z < h_2. \tag{50}$$

Note that these boundary conditions correspond to the continuity of oxygen concentration and mass flux at the interface between regions.

Substituting the inner and outer solutions into (48) gives

$$
\phi_0 + \sum_{n=1}^{\infty} A_n (K_0'(\omega_n ro) I_0(\omega_n r_T) \quad -I_0'(\omega_n ro) K_0(\omega_n r_T)) \cos(\omega_n z)
$$
$$
= U_{II}(z) + \sum_{n=1}^{\infty} \alpha_n I_0(\sqrt{\lambda_n} r_T) Z_n^{(2)}(z).
\tag{51}
$$

Multiplying both sides of this equation by $Z_m^{(2)}(z)$, $m = 1, 2, \ldots$ and integrating between $z = 0$ and $z = H$ allows exploitation of the orthogonality of the eigenfunctions and the system

$$
RA = T\boldsymbol{\alpha} + \boldsymbol{g}
\tag{52}
$$

is obtained. Further substituting the inner and outer solutions into (49) and (50), multiplying by $Z_m^{(2)}(z)$, $m = 1, 2, \ldots$, integrating between $z = 0$ and $z = H$ and finally adding all three equations together gives

$$
SA = W\boldsymbol{\alpha},
\tag{53}
$$

where the matrices and vectors appearing above have entries

$$
R_{mn} = (K_0'(\omega_n ro) I_0(\omega_n r_T) - I_0'(\omega_n ro) K_0(\omega_n r_T)) \int_0^H Z_m^{(2)} Z_n^{(1)} \, dz,
\tag{54}
$$

$$
S_{mn} = \omega_n (K_0'(\omega_n ro) I_0'(\omega_n r_T) - I_0'(\omega_n ro) K_0'(\omega_n r_T)) \int_0^H Z_m^{(2)} Z_n^{(1)} \, dz,
\tag{55}
$$

$$
T_{mn} = I_0(\sqrt{\lambda_m} r_T) \int_0^H Z_m^{(2)} Z_n^{(1)} \, dz,
\tag{56}
$$

$$
W_{mn} = \sqrt{\lambda_m} I_0'(\sqrt{\lambda_m} r_T) \int_0^H Z_m^{(2)} Z_n^{(1)} \, dz,
\tag{57}
$$

$$
g_m = \int_0^H (\phi_0 - U_{II}(z)) Z_m^{(2)} \, dz,
\tag{58}
$$

for $m, n = 1, 2, \ldots$ and $A = (A_1, A_2, \ldots)^T$ and $\boldsymbol{\alpha} = (\alpha_1, \alpha_2, \ldots)^T$. Combining (52) and (53) yields the expressions

$$
\boldsymbol{\alpha} = (T - RS^{-1}W)^{-1}\boldsymbol{g},
\tag{59}
$$

$$
A = (TW^{-1}S - R)^{-1}\boldsymbol{g}.
\tag{60}
$$

## Parameter estimation

For numerical simulations, the values of model parameters were estimated. Experimentally, the PCLS is suspended in 1.3ml of William E Media, in a single well of a 12-well plate. The well is 17.5 mm deep, with a diameter of 22.1mm, yielding a media depth of approximately 3.4mm.

A summary of parameters used in the model can be found in Table 1. Briefly, the remaining parameters in the problem are derived from the literature. The oxygen transport parameters within the media and the slice are taken to be $4.85 \times 10^{-9} \text{m}^2/\text{s}$ and $1.6 \times 10^{-9} \text{m}^2/\text{s}$ respectively,

**Table 1. The parameter values used within the numerical simulations.**

| Parameter | Description | Value |
|---|---|---|
| $H$ | Depth of fluid | 3.4mm |
| $h_2 - h_1$ | Thickness of liver slice | $2.5 \times 10^{-4}$m |
| $r_T$ | Radius of the liver slice | 2.5mm—5.0mm diameter |
| | | 4.0mm—8.0mm diameter |
| $r_o$ | Radius of the well | 1.1mm |
| $D_1$ | Oxygen diffusion rate within the slice | $1.6 \times 10^{-9}$m$^2$/s |
| $D_2$ | Oxygen diffusion rate within the media | $4.85 \times 10^{-9}$m$^2$/s |
| V | Rate of oxygen uptake within the tissue | $0.057$s$^{-1}$ |

as these are the parameters used by [15] in an investigation into oxygen diffusion within hepatic spheroids. The oxygen uptake rate is assumed to be $0.057s^{-1}$ as this is the value used by [16] in their investigation into oxygen consumption in liver tissue. Finally, it is assumed that atmospheric oxygen pressure is 160mmHg.

## Results

Two PCLS diameters were considered in this study, 5mm and 8mm, with a thickness of $250\mu m$ [7, 17]. The internal oxygen concentrations of each diameter were investigated at atmospheric 21%, 80% and 95% external incubator oxygen concentration, yielding a total of 6 simulated experimental scenarios.

## Numerical implementation

The solutions $\phi_I$ and $\phi_{II}$ given by (47), (35), (59) and (60) were coded into Matlab R2020b to produce numerical solutions. As the solutions of $\phi$ contain infinite summations, they must be truncated to contain a maximum of $M$ terms. This makes the computational complexity of the solution O($M^3$), as Gaussian elimination is used to solve (59) and (60) [18].

After investigation (not shown here), it was found that setting $M = 5$ is sufficient to capture an accurate solution within the media, and setting $M = 7$ captures an accurate solution within the tissue. Summing a greater number of terms in both cases does not lead to a visible change in the results.

## Model predictions

Initial simulations predict the internal steady state oxygen concentration as a function of where the PCLS is located in the well. Three scenarios were examined, i) the tissue resting at the bottom of the well, ii) the centre of the tissue positioned in the middle of the well, and iii) the top of the tissue positioned one micron below the surface of the media. (Table 2 specifies the values of $h_1$ and $h_2$ used in these simulations. Figs (2)–(4) present contour plots of the

**Table 2. The values of $h_1$ and $h_2$ that describe the three different locations of the PCLS.**

| Position of PCLS | $h_1(mm)$ | $h_2(mm)$ |
|---|---|---|
| Bottom of well | 0 | 0.25 |
| Middle of well | 1.575 | 1.825 |
| Top of Well | 3.149 | 3.399 |

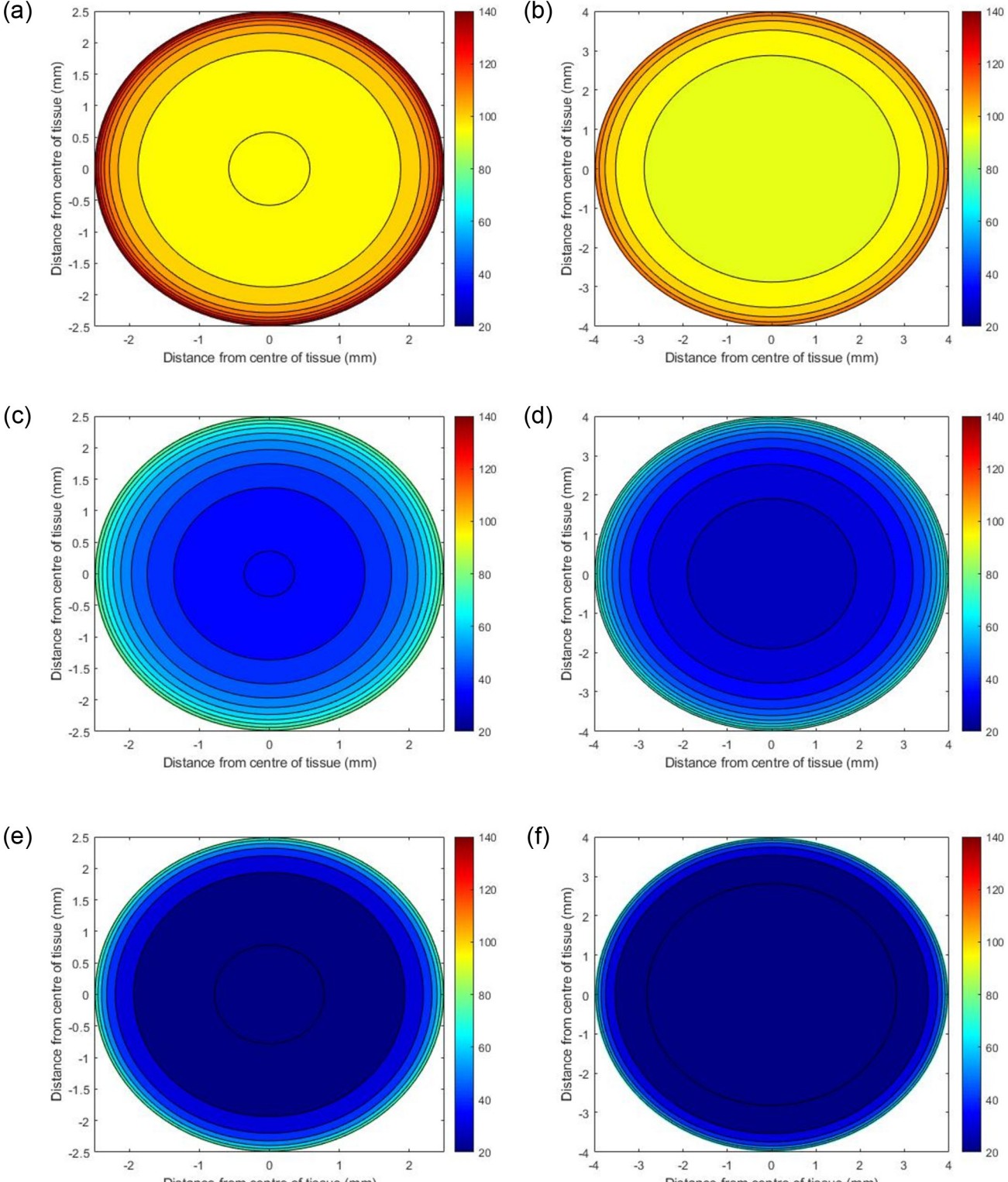

**Fig 2. Predicted oxygen concentrations through the centre of PCLS under atmospheric oxygen conditions (21%) with different diameters.** The panel on the left shows the concentrations of oxygen when the PCLS is at the top of the well (a), the middle of the well (c) and the bottom of the well (e) for a slice with 5mm diameter, whilst (b), (d) and (f) depict the concentrations in the same locations for a slice with 8mm diameter.

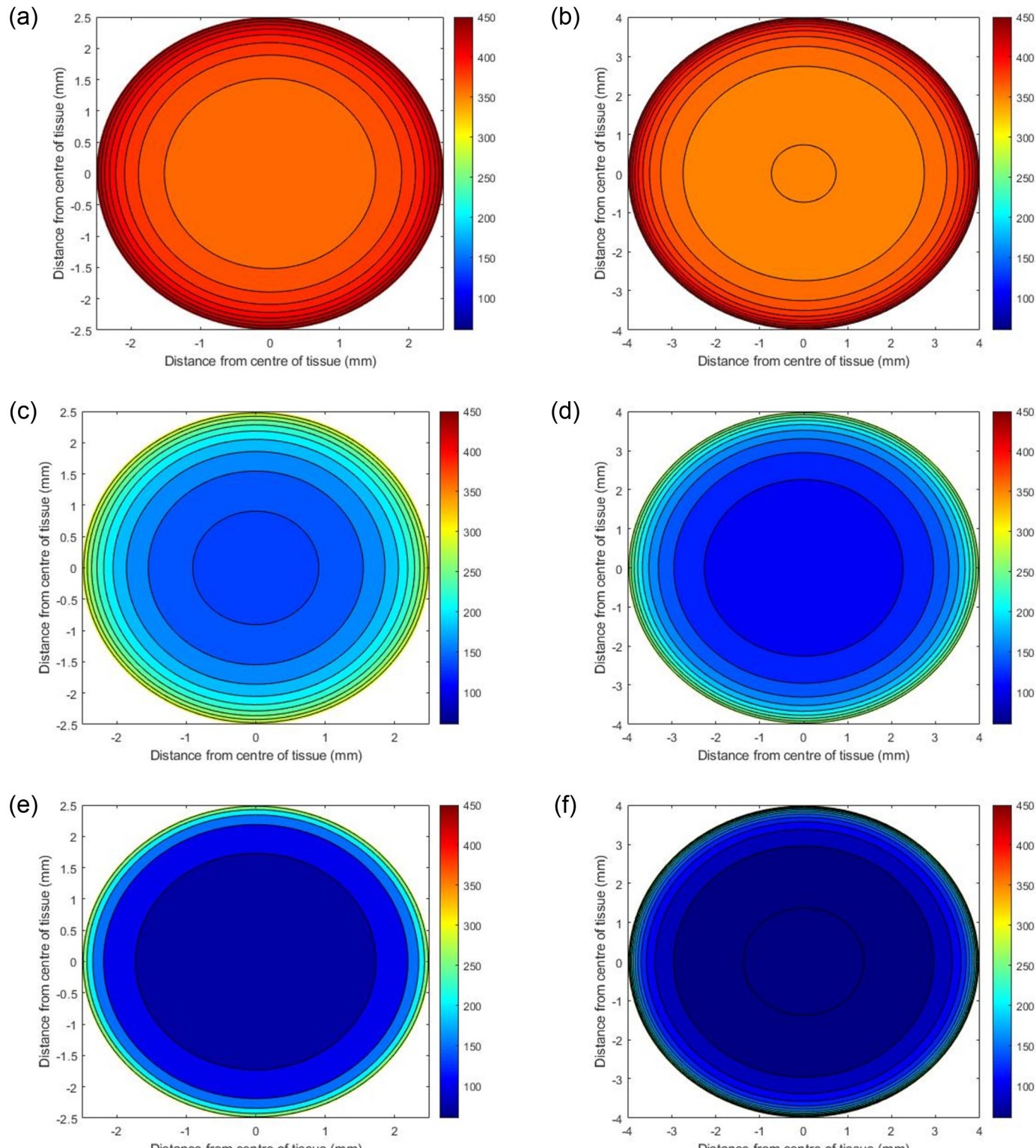

**Fig 3. Predicted oxygen concentrations through the centre of PCLS under 80% oxygen with different diameters.** The panel on the left shows the concentrations of oxygen when the PCLS is at the top of the well (a), the middle of the well (c) and the bottom of the well (e) for a slice with 5mm diameter, whilst (b), (d) and (f) depict the concentrations in the same locations for a slice with 8mm diameter.

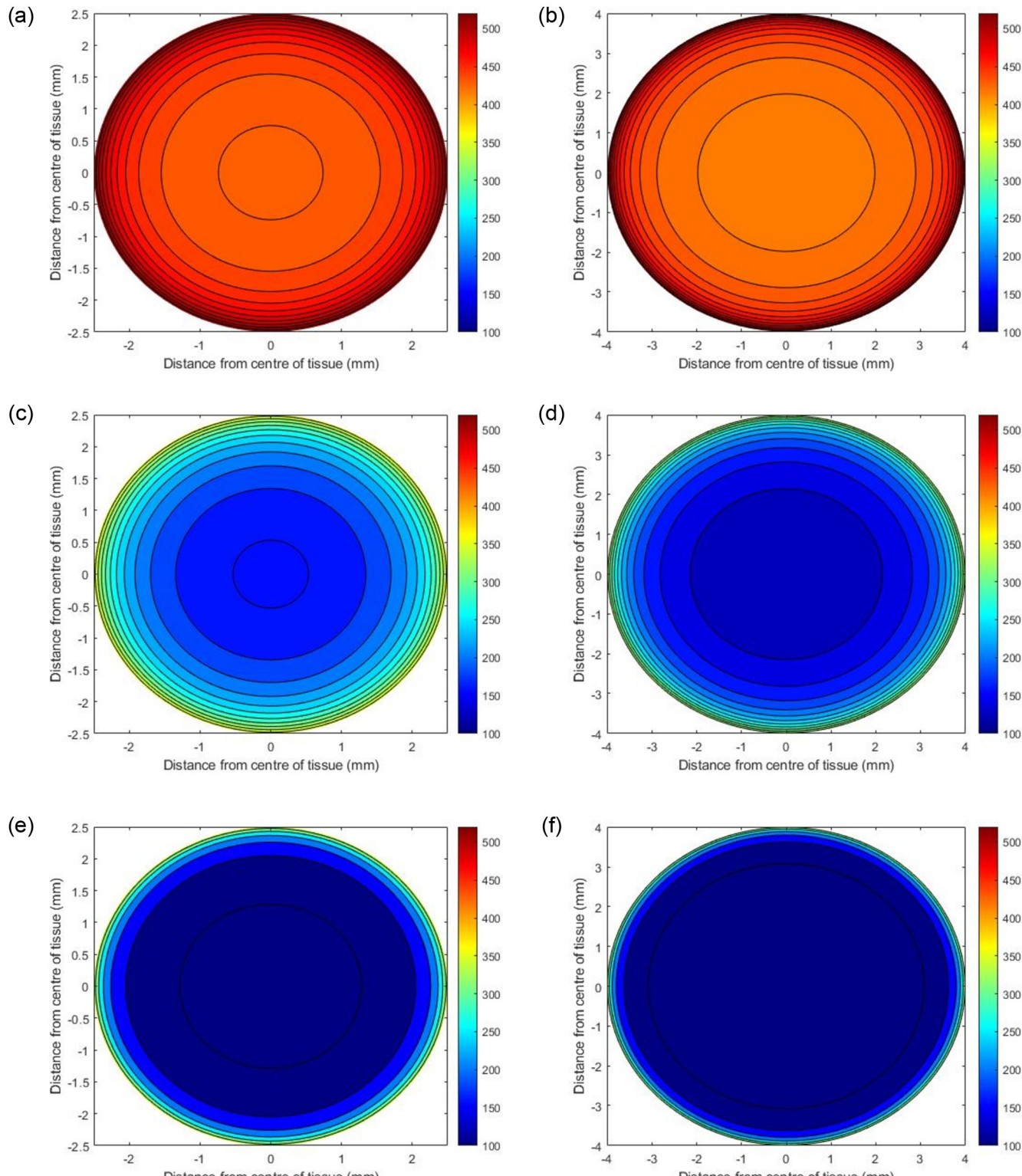

**Fig 4. Predicted oxygen concentrations through the centre of PCLS under 95% oxygen with different diameters.** The panel on the left shows the concentrations of oxygen when the PCLS is at the top of the well (a), the middle of the well (c) and the bottom of the well (e) for a slice with 5mm diameter, whilst (b), (d) and (f) depict the concentrations in the same locations for a slice with 8mm diameter.

oxygen concentration through the cross section of the centre of the slice ($z = \frac{1}{2}(h_1 + h_2)$) at different well depths for PCLS of both diameters.

**Effects of well depth.** Figs 2–4 illustrate how the oxygen concentration at the center of the slice changes as a function of where it is positioned in the well at atmospheric, 80% and 95% oxygen respectively. The highest internal oxygen concentrations in each simulation arise when the PCLS is located at the top of the well, i.e. $1\mu m$ from the media/air interface. In each simulation, the concentration of oxygen at the centre of the slice decreases as the tissue moves to the bottom of the well.

**Effects of incubator oxygen concentration.** The concentration of oxygen in the incubator induces a significant effect on the oxygen concentration at the centre of all slices at all depths of the well. In general, the higher the incubator concentration, the higher the internal PCLS oxygen concentration. 95% and 80% incubator oxygen concentrations yield supraphyisiological concentrations of oxygen across the centre of the slice. Under 95% oxygen conditions, the oxygen concentration can range from 100–520mmHg for both diameters, depending upon the depth the tissue is submerged in the well. For 80% conditions, the ranges are 80–440mmHg and 60–440mmHg (5mm and 8mm diameter), whereas at atmospheric oxygen conditions, the concentration of oxygen at the centre of the slice ranges from 20–140mmHg for both diameters, again, depending upon the depth of the tissue in the well.

**Effects of PCLS diameter size.** In all cases, the model predicts that the edges of the PCLS, for both 5mm and 8mm tissues, are the most oxygenated sections, with the concentration decreasing as the distance from the centre of the slice increases. There are notable differences between the 5mm and 8mm slices, with simulations predicting that the smaller 5mm slices are more oxygenated internally at steady state. This is apparent for all slices, in all experimental simulations.

## Variation of oxygen concentration through the PCLS

Fig 5 shows the variation in oxygen concentration through the thickness of the PCLS of two different radii at atmospheric conditions. In both figures, the blue line corresponds to the depth within the well that yields a maximum concentration of 65mmHg, while the red line corresponds to the depth that yields a minimum concentration of 35mmHg. For both the 5mm and 8mm PCLS, the bottom of the tissue is more oxygenated than the centre of the tissue, while the top of the tissue is the most oxygenated. This follows that the centre of the tissue is not directly exposed to oxygen in the media, while the bottom and top of the tissue are. Indeed, the top of the tissue ($250\mu m$) is most oxygenated. Fig 5 also shows the difference in oxygenation tolerance between a 5mm and 8mm slice, with the 8mm slice having a smaller difference in concentration through the slice compared to the 5mm PCLS. This finding is exemplified by the results in Table 3, which shows the position in the well that the PCLS of different radii must occupy to be at physiological concentrations of oxygen. Table 3 quantifies how fine this margin of $h_1$ and $h_2$ is for obtaining physiologically-relevant oxygen concentrations.

**Simulating factors effecting the minimum concentration of oxygen in the tissue.** The model was used to further explore how the minimum and maximum concentration of oxygen changes in a 5mm PCLS at atmospheric conditions from the top to the bottom of the well. Fig 6 shows model predictions for the maximum (blue line) and minimum (red line) value of oxygen concentration within the tissue as the tissue descends from the top to the bottom of the well.

The model predicts that as the PCLS descends from the surface to the bottom of the well, both the maximum and minimum oxygen concentration in the slice decreases. Simulations suggest that the oxygen concentration (minimum and maximum) decreases the most as the

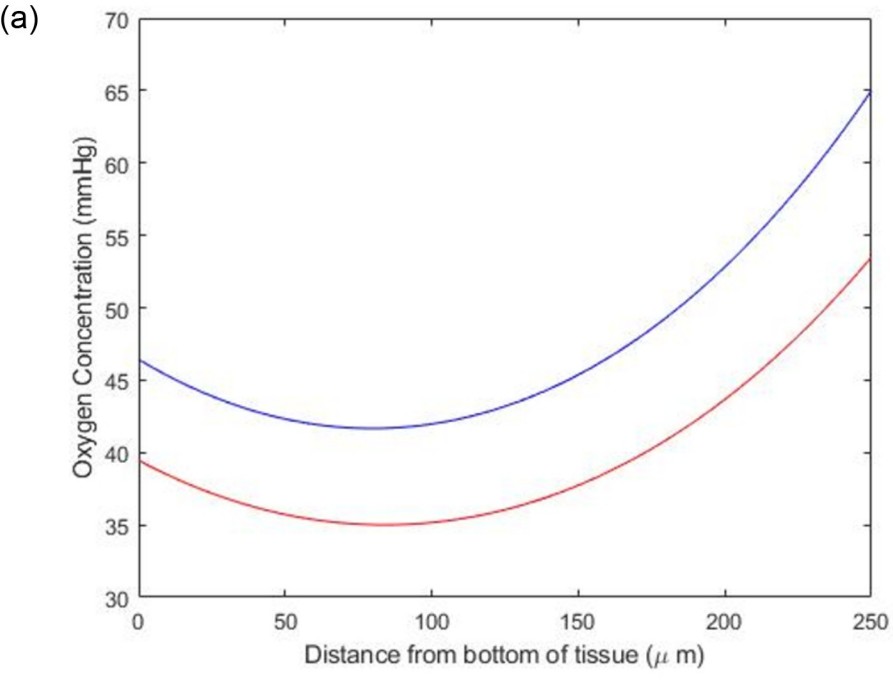

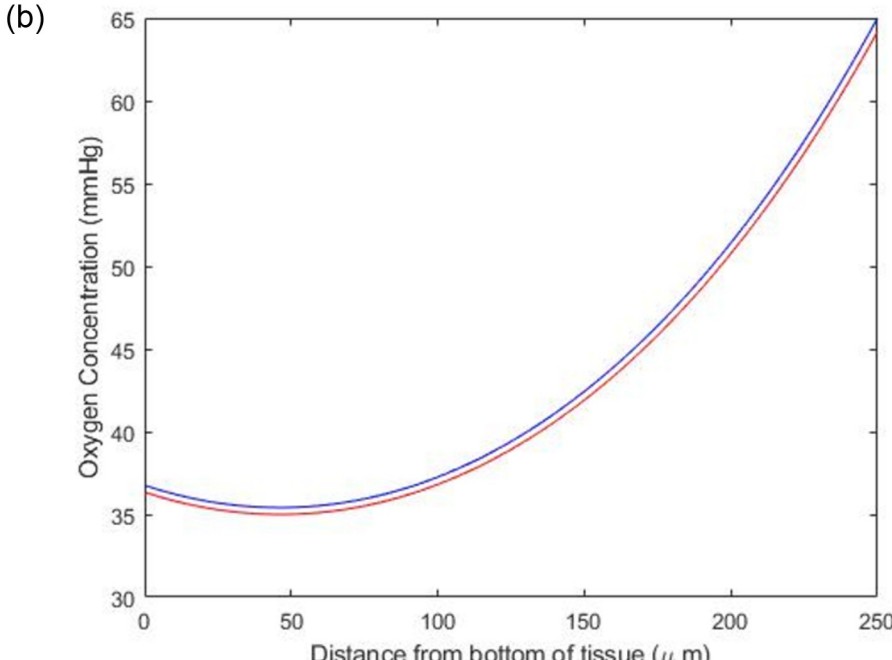

**Fig 5. The variation of oxygen concentration through the thickness of a) a 5mm and b) an 8mm PCLS at 21% oxygen concentration.** The blue line corresponds to the depth within the well that gives a maximum concentration of 65mmHg, and the red line corresponds to the depth within the well (in a 12-well plate) that gives a minimum concentration of 35mmHg.

**Table 3. The position within the well that PCLS of different radii must occupy to be at physiological concentrations of oxygen. The values in this table have been used to create Fig 5.**

| Tissue Radius | Concentration | $h_1$ (mm) | $h_2$ (mm) |
|---|---|---|---|
| 5mm | Maximum 65mmHg | 2.1531 | 2.4031 |
| | Minimum 35mmHg | 1.6881 | 1.9381 |
| 8mm | Maximum 65mmHg | 2.2614 | 2.5114 |
| | Minimum 35mmHg | 2.2419 | 2.4919 |

slice descends from 0 to 1mm below the surface of the well, with a more gradual oxygen concentration drop-off for the remaining 2mm descent. It should be noted that the predicted position of the PCLS to yield physiologically relevant oxygen concentrations (Table 3) lie within the less-sensitive 1–3mm region of the well.

Fig 7 investigates how the minimum concentration of oxygen changes as a function of the amount of media in the well, assuming that the PCLS is ($1\mu m$) below the surface of the well. Simulations suggest that initially, the minimum oxygen concentration in the PCLS increases as the volume of media in which it is suspended increases from 0 to 1.5ml. However, the model predicts that as the volume of the media increases from 0.5ml to 1.5ml, the minimum oxygen concentration increases at a significantly reduced rate. Eventually, the addition of media into the well becomes counter-productive and the minimum oxygen concentration in the tissue begins to decrease.

## Discussion

The aim of this study was to develop a mathematical model to predict the optimal experimental conditions that yield a physiologically-relevant internal oxygen concentration in a precision cut liver slice. In general, simulations explored a range of plausible 12-well experimental

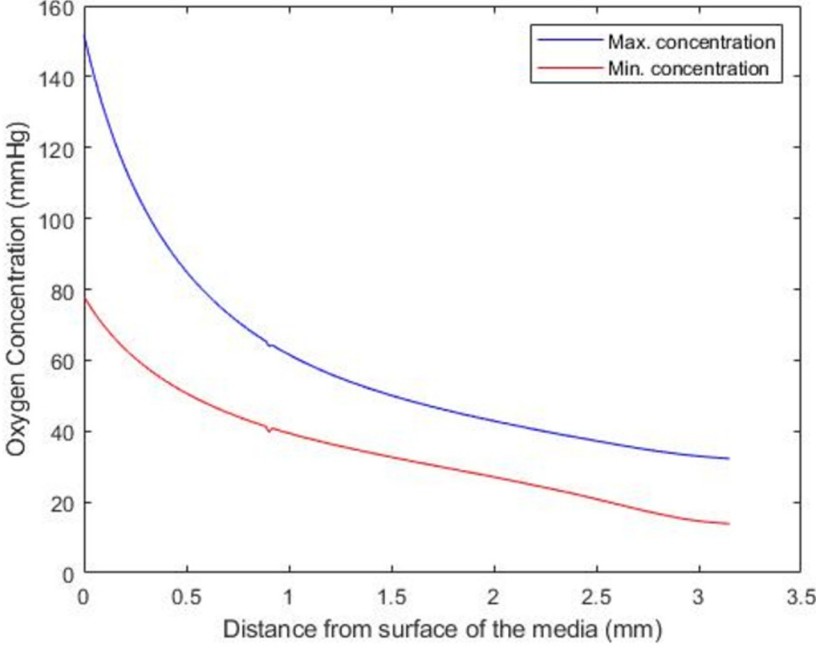

**Fig 6. The minimum (red line) and maximum (blue line) value of oxygen concentration within the tissue as the position of the PCLS is moved from the surface of the well to the bottom of the well.**

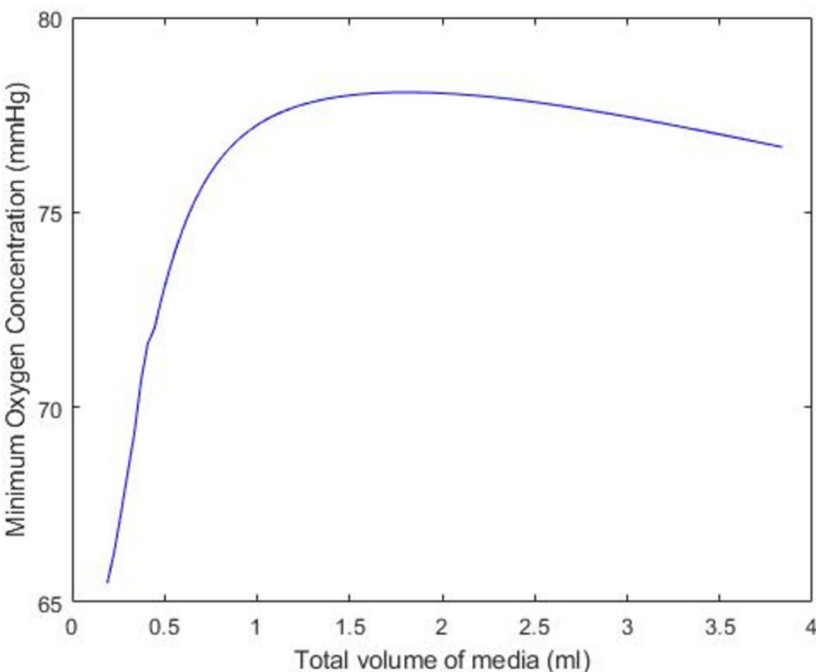

**Fig 7. The minimum value of oxygen within the PCLS as the experimental volume of the media changes.** The PCLS is assumed to lie $1\mu m$ below the surface of the media, for a 5mm slice at atmospheric oxygen in a 12-well plate.

scenarios based on current methods deployed in multiple laboratories. Specifically, simulations investigate how alterations in external oxygen concentration, slice diameter and position in the well effect internal oxygen concentration.

For all three external oxygen concentrations, simulations suggest that internal oxygen concentration is sensitive to the position of the slice within the well (Figs 2–4). Unsurprisingly, the closer to the media-oxygen interface, the higher the oxygen concentration in the PCLS. These simulations also predict that for the larger 8mm diameter slice, the centre of the slice is more deoxygenated compared to the smaller 5mm slice. Having experimental control of internal oxygen concentrations when using 3D *in vitro*, is crucial, as hypoxia is one of the possible factors limiting functional longevity of these systems [1, 8, 19]. With respect to identifying which experimental setup will yield physiologically-relevant internal oxygen concentrations of 35–65mmHg for the liver [7], simulations predict that elevated external oxygen concentrations of 80 and 95% oxygen generates supraphysiological concentrations of oxygen within the PCLS, which is in good accordance with the literature [20]. Model predictions further indicate that using atmospheric oxygen would deliver the most representative internal oxygen concentration, depending on the position of the slice within the well, which is also in good accordance with the literature [8].

Other studies have reported that while internal slice oxygen concentration is also sensitive to the thickness of the slice, a balance must be struck between using thinner slices that have lower viability but higher oxygen permeability, and thicker slices that sacrifice internal oxygen concentration control for experimental quality and viability.

Having predicted that setting the incubator to atmospheric oxygen concentration, Fig 5 compares the required depths necessary to produce a maximum concentration of 65mmHg (blue line) and a minimum concentration of 35mmHg (red line) for a 5mm and 8mm PCLS at atmospheric oxygen. Simulations suggest that while both diameters are capable of having the

correct oxygen concentration, the 5mm diameter yields a higher degree of separation. This finding is confirmed in Table 3, which shows that the tolerance for the precise predicted position in the well required to generate the physiological concentrations of oxygen is significantly smaller for the 8mm than the 5mm diameter. In real terms, this simulation would prompt the use of the 5mm slice on the basis of accuracy and reproducibility of experimental setup.

Having honed in on the predicted optimal conditions, the final two simulations show how the minimum and maximum concentration within the tissue changes as the tissue position moves from the surface of the well to the bottom (Fig 6), followed by how sensitive the minimum oxygen concentration in the tissue is to the volume of media in the well (Fig 7). While the correlation between depth and deoxygenation is unsurprising, (the deeper the slice, the more deoxygenation) (Fig 6), the relationship between media volume and the minimum concentration of oxygen in the PCLS is interesting. Fig 7 suggests that to create reproducible oxygen concentrations, the volume of media must be consistent and defined. Using volumes of 1.5ml or less could result in variability in results as a function of the sensitivity to the media volume in this range.

It is commonplace to set the PCLS at the bottom of the well, similar to the majority of *in vitro* cell culture and use an agitation method causing the slice to artificially and unpredictably be 'suspended'. The position of the slice within the well can be more accurately manipulated with the use of an insert, at greater financial cost. Here, the model assumes that the insert does not interfere with oxygen diffusion on the underside of the slice.

## Conclusion

PCLS provide a more physiologically relevant basis for studying liver disease compared to *in vitro* methods. However, their use is severely limited by experimental variability and slice viability. This study aimed to address whether PCLS viability as a function of its internal oxygen concentration could be improved using mathematical modelling. Simulations explored a range of external incubator oxygen concentrations, different slice diameters, and different positions within the well of a 12-well plate. The model predicts that an internal oxygen gradient of 35–65mmHg, comparable to that within the liver, is achievable across a 5mm diameter liver slice at atmospheric oxygen, provided that the position within the well satisfies a maximum height of between 2.15mm and 2.40mm, and a minimum height of 1.69mm and 1.94mm. The ability to generate a physiologically relevant internal oxygen concentration at atmospheric oxygen concentrations is experimentally encouraging and could improve experimental replicability. We plan to use these optimised parameters experimentally to determine if these model predictions can improve slice viability and longevity in culture. Using mathematical modelling in this manner will help to reduce the number of specimens required to validate the PCLS model, reducing reagent/ consumable costs and researcher time. Furthermore, the proposed methodology could be deployed to guide the design of alternative experiments, for example, using 24 or 96-well plates rather than 12-well plates. Improving the relevance and viability of using PCLS not only ameliorates mechanistic understanding of liver diseases, but it also potentially reduces the need for the use of animal models.

The mathematical model developed in this work will prove invaluable in guiding the direction of future experimental work, and the authors intend to conduct investigations of this type in the near future. However, some of the underlying assumptions used in the model derivation are relatively simple. The model assumes that oxygen absorption within the tissue is proportional to the oxygen concentration within the PCLS, whereas other studies have suggested that a non-linear model may be a more appropriate model for this process. Furthermore, a steady state solution of oxygen concentration within the PCLS has been obtained,

whereas a time-dependent model would provide a greater indication of how long it takes to obtain the desirable physiologically relevant conditions. Our future work will seek to address some of these limitations by creating refined mathematical models.

## Supporting information

**S1 Data.**
(PDF)

## Author Contributions

**Conceptualization:** S. J. Chidlow, L. E. Randle, R. A. Kelly.

**Formal analysis:** S. J. Chidlow, R. A. Kelly.

**Investigation:** S. J. Chidlow, R. A. Kelly.

**Methodology:** S. J. Chidlow.

**Software:** S. J. Chidlow.

**Writing – original draft:** S. J. Chidlow, L. E. Randle, R. A. Kelly.

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
