## [Decision Letter · Decision Letter 0]

16 Aug 2022

PONE-D-22-21694Predicting physiologically-relevant oxygen concentrations in precision-cut liver slices using mathematical modellingPLOS ONE

Dear Dr. Chidlow,

Thank you for submitting your manuscript to PLOS ONE. After careful consideration, we feel that it has merit but does not fully meet PLOS ONE’s publication criteria as it currently stands. Therefore, we invite you to submit a revised version of the manuscript that addresses the points raised during the review process.

We look forward to receiving your revised manuscript.

Kind regards,

Ashwani Kumar, Ph.D.

Academic Editor

PLOS ONE

Journal Requirements:

2. Please note that PLOS ONE has specific guidelines on code sharing for submissions in which author-generated code underpins the findings in the manuscript. In these cases, all author-generated code must be made available without restrictions upon publication of the work. Please review our guidelines at https://journals.plos.org/plosone/s/materials-and-software-sharing#loc-sharing-code and ensure that your code is shared in a way that follows best practice and facilitates reproducibility and reuse. New software must comply with the Open Source Definition.

Reviewers' comments:

Reviewer's Responses to Questions

**Comments to the Author**

1. Is the manuscript technically sound, and do the data support the conclusions?

Reviewer #1: Yes

Reviewer #2: Yes

2. Has the statistical analysis been performed appropriately and rigorously? 

Reviewer #1: Yes

Reviewer #2: Yes

3. Have the authors made all data underlying the findings in their manuscript fully available?

Reviewer #1: Yes

Reviewer #2: Yes

4. Is the manuscript presented in an intelligible fashion and written in standard English?

Reviewer #1: Yes

Reviewer #2: Yes

5. Review Comments to the Author

Reviewer #1: Dear Authors

I have now completed the review of the manuscript titled "Predicting physiologically-relevant oxygen concentrations in precision-cut liver slices using mathematical modelling". The aim of this study was to predict whether it is possible to generate a

physiologically relevant oxygen gradient of 35-65mmHg across a precision cut liver slice

using mathematical modelling. Simulations explore how the internal oxygen

concentration changes as a function of the diameter of the slice, the position inside the

well and the external incubator oxygen concentration. The model predicts that the

desired oxygen gradient may be achieved using a 5mm diameter slice at atmospheric

oxygen concentrations, provided that the slice is positioned at a certain height within

the well of a 12-well plate. Overall the paper is very interesting however it requires to incorporate some suggestions to further improve the quality of the manuscript.

1. Introduction section requires a new sub section where author introduced latest AI and ML applications in different areas [1-11], including mathematical modelling for medical applications, Please add these in the new sub section section.

2. Please provide the computational complexity of the developed models with their machine information, authors can check and use these studies CDLSTM, SMOTEDNN, DNNBOT, PCCNN etc.

4. What is the future scope of the proposed research, authors should describe the limitations in good way. After introducing AI/ML models and showing implementation of their mathematical model the future scope can be determined.

5. These days ML and AI are utilized to solve several applications which are based on different parameters, I suggest to make a small paragraph which discusses the role of AI and ML methods, authors can use some of the references provided in the comments 1 and 2 and then justify the use of their mathematical model.

7. Conclusion require more information on pros, cons, future scope.

References

1. Artificial neural network‐based modeling of snow properties using field data and hyperspectral imagery

2. SMOTEDNN: A Novel Model for Air Pollution Forecasting and AQI Classification

3. CDLSTM: A Novel Model for Climate Change Forecasting

4. CNN Based Automated Weed Detection System Using UAV Imagery

5. Assessment of trends of land surface vegetation distribution, snow cover and temperature over entire Himachal Pradesh using MODIS datasets

6. Analysis of environmental factors using AI and ML methods

7. Machine Learning-based Classification of Hyperspectral Imagery

8. Fusion-Based Deep Learning Model for Hyperspectral Images Classification

9. Snow and glacial feature identification using hyperion dataset and machine learning algorithms

10. Assessment of trends of land surface vegetation distribution, snow cover and temperature over entire Himachal Pradesh using MODIS datasets

11. Deep Learning Based Modeling of Groundwater Storage Change

Reviewer #2: This paper is an interesting paper, tackling the problem of predicting physiologically-relevant oxygen concentrations in precision-cut liver slices. Mathematical modeling, specifically simulations explored external incubator oxygen concentrations, different slice diameter, and different positions within the well of a 12-well plate.

The model predicts 35-65 mmHg internal oxygen gradient is possible across liver slice at atmospheric oxygen, with some provided constraints. This is a surprising and encouraging result. This work can also guide the design of alternative experiments.

Besides, the paper is well written, with a clear explanation of the formula used in the design.

6. PLOS authors have the option to publish the peer review history of their article (what does this mean?). If published, this will include your full peer review and any attached files.

Reviewer #1: **Yes: **Mohd Anul Haq

Reviewer #2: No

---

## [Author Response · Author response to Decision Letter 0]

20 Sep 2022

Dear Dr. Kumar

We wish to thank our reviewers for the exceptionally fast time in which they reviewed our manuscript, and for their constructive feedback. Please find our responses to the points raised below:

Reviewer 1:

1) Machine learning is a widely used tool in many different application areas but does not seem to have been used in many investigations relating to precision cut liver slices (PCLS) or similar. We have performed a wider literature search in this area and were only able to find one paper that uses machine learning (see below).

Abdeltawab H, Khalifa F, Hammouda K, Miller JM, Meki MM, Ou Q, et al. “Artificial Intelligence Based Framework to Quantify the Cardiomyocyte Structural Integrity in Heart Slices”. Cardiovascular Engineering and Technology. 2022;13(1):170–180

This appears as reference [14] in the revised version of the manuscript and is discussed briefly in the penultimate paragraph of the Introduction.

2) The largest computational expense of our solution is solving the linear system of M equations given by (60) and (61), where M≥1. As this system is solved using Gaussian elimination, the computational complexity of the solution is O(M^3 ). We always choose M≤7, so it takes Matlab approximately 1 second to compute and plot the solution. This information has been included in the section entitled “Numerical Implementation”, and a new reference relating to computational complexity has been provided as reference [18]. Details of the reference are below:

Arora S, Barak B. Computational complexity: a modern approach. Cambridge University Press; 2009

3) We have included some information about the limitations of our model and plans for future work in the “Conclusions” at the end of the paper. Our planned future work includes:

 *Conducting experimental work on PCLS guided by the model derived in this work to generate data for future 

 computational models.

 *Solving a time-dependent model of oxygen diffusion and absorption to estimate how oxygen concentrations 

 in PCLS change over time.

4) We agree that machine learning algorithms could prove to be a valuable tool in this area to help predict optimal positioning of a PCLS within a well to achieve physiologically realistic concentrations. However, the available literature in this area does not contain enough data to make the use of machine learning algorithms viable, and we do not currently have enough experimental data of our own to use either. We hope that this area will become increasingly data-rich over the coming years and make the use of these techniques feasible, but at the moment, the only possible mathematical models to use are deterministic, which is why we chose this particular approach.

As stated above in 1., we have included a small section about applications of machine learning in a similar area.

5) Please see 3. above for how we have responded to this point.

Reviewer 2:

No changes to the manuscript were requested, so we have no points to rebut. We thank you very much for your feedback and review.

Thank you for further consideration of our manuscript. We hope that these amendments are sufficient.

Yours sincerely,

Stewart Chidlow, Laura Randle and Ross Kelly.

---

## [Decision Letter · Decision Letter 1]

26 Sep 2022

Predicting physiologically-relevant oxygen concentrations in precision-cut liver slices using mathematical modelling

PONE-D-22-21694R1

Dear Dr. Chidlow,

We’re pleased to inform you that your manuscript has been judged scientifically suitable for publication and will be formally accepted for publication once it meets all outstanding technical requirements.

Kind regards,

Ashwani Kumar, Ph.D.

Academic Editor

PLOS ONE

Additional Editor Comments (optional):

Reviewers' comments:

Reviewer's Responses to Questions

**Comments to the Author**

1. If the authors have adequately addressed your comments raised in a previous round of review and you feel that this manuscript is now acceptable for publication, you may indicate that here to bypass the “Comments to the Author” section, enter your conflict of interest statement in the “Confidential to Editor” section, and submit your "Accept" recommendation.

Reviewer #1: All comments have been addressed

Reviewer #2: All comments have been addressed

2. Is the manuscript technically sound, and do the data support the conclusions?

Reviewer #1: Yes

Reviewer #2: Yes

3. Has the statistical analysis been performed appropriately and rigorously? 

Reviewer #1: Yes

Reviewer #2: Yes

4. Have the authors made all data underlying the findings in their manuscript fully available?

Reviewer #1: Yes

Reviewer #2: Yes

5. Is the manuscript presented in an intelligible fashion and written in standard English?

Reviewer #1: Yes

Reviewer #2: Yes

6. Review Comments to the Author

Reviewer #1: I have now completed the review of the revised manuscript titled " Predicting physiologically-relevant oxygen concentrations in precision-cut liver slices using mathematical modelling”. I have observed that the authors put good efforts to address all the comments satisfactorily.

Reviewer #2: I am glad the authors revised the paper and addressed the reviewer comments in an elegant way. Predicting physiologically-relevant oxygen concentrations in liver slices is interesting, and has both scientific and society importance.

7. PLOS authors have the option to publish the peer review history of their article (what does this mean?). If published, this will include your full peer review and any attached files.

Reviewer #1: **Yes: **Mohd Anul Haq

Reviewer #2: No

---

## [Editor Report · Acceptance letter]

30 Sep 2022

PONE-D-22-21694R1 

Predicting physiologically-relevant oxygen concentrations in precision-cut liver slices using mathematical modelling 

Dear Dr. Chidlow:

I'm pleased to inform you that your manuscript has been deemed suitable for publication in PLOS ONE. Congratulations! Your manuscript is now with our production department. 

Kind regards, 

on behalf of

Dr. Ashwani Kumar 

Academic Editor

PLOS ONE